# Corrosion Resistance of Mn-Containing AFA Alloys in LBE at 550 °C with Controlled Oxygen Concentration of 10^−6^ wt.%

**DOI:** 10.3390/ma18061328

**Published:** 2025-03-17

**Authors:** Menghe Tu, Yajie He, Zihui Liu, Xiaogang Fu, Lingzhi Chen

**Affiliations:** 1Reactor Engineering Technology Research Department, China Institute of Atomic Energy, Xinzhen, Fangshan District, Beijing 102413, China; tumenghe11@tsinghua.org.cn (M.T.); heyajie_stu@163.com (Y.H.); llzihui@163.com (Z.L.); 2China Nuclear Power Engineering Co., Ltd., Beijing 100840, China

**Keywords:** LBE, corrosion, AFA steel, Mn element

## Abstract

Alumina-forming austenitic steels (AFA steels) exhibit excellent creep resistance and oxidation capabilities, making them a strong candidate for cladding materials in lead-cooled fast reactors. This study investigates the corrosion resistance of Mn-containing AFA steels in lead–bismuth eutectic (LBE) at 550 °C with a controlled oxygen concentration of 10^−6^ wt.%. The results demonstrate that under these experimental conditions, the addition of Al enhances the material’s resistance to lead–bismuth corrosion. Moreover, Mn incorporation significantly improves corrosion resistance, with the optimal composition being an AFA alloy containing 16 wt.% Ni, 12 wt.% Cr, 3 wt.% Al, and 4 wt.% Mn. Mn addition alters the type of oxide product formed on the alloy surface, shifting from Fe_3_O_4_ or (Fe, Cr)_x_O_y_ to (Cr, Mn)_x_O_y_.

## 1. Introduction

The lead–bismuth eutectic (LBE) coolant possesses excellent neutronic, thermohydraulic, chemical inertness, and inherent safety properties, which make it highly promising for applications in lead-cooled reactors, accelerator-driven subcritical nuclear energy systems (ADS), and fusion reactors [1]. However, the severe corrosion of structural materials caused by liquid metal LBE in lead-cooled reactors poses significant challenges. Conventional commercial austenitic stainless steels (e.g., 316L, 304L, and 15-5Ti) and ferritic/martensitic steels (e.g., T91, HT9, P22, and EP823) fail to meet the service performance requirements under the high-temperature operational conditions of future lead-cooled reactors.

The severity of corrosion in lead-cooled reactors lies in the co-occurrence of dissolution and oxidation corrosion of structural materials in liquid LBE. Research by Valentyn Tsisar et al. [2,3] on austenitic steels (1.4970, 316L, and 1.4571) in 550 °C LBE with an oxygen concentration of 10^−6^ wt.% shows that although oxidation dominates, dissolution corrosion caused by selective leaching of elements remains significant. Over time, beneath the initially formed oxide layer composed of α-Cr_2_O_3_ and spinel (Fe,Cr)_x_O_y_, Pb and Bi penetration is also observed, indicating that Cr-containing oxide layers at elevated temperatures are insufficient to resist LBE corrosion. Developing new materials with excellent resistance to high-temperature LBE corrosion is one of the critical challenges for enabling the engineering application of lead-cooled reactors [4].

The standard Gibbs free energy of formation for Al oxides is lower compared to Fe, Cr, and Si, allowing for the formation of a dense and thermodynamically stable Al_2_O_3_ film. Alumina-forming austenitic (AFA) alloys were proposed by Oak Ridge National Laboratory in 2007 [5,6]. These alloys are novel materials developed by adding Al and Nb to ultrafine-grained steels. The bulk contains numerous fine precipitates that enhance the material’s strength and resistance to radiation damage. AFA alloys exhibit excellent corrosion resistance and high-temperature creep resistance.

Research on the application of AFA alloys in lead-cooled reactors remains relatively limited. Ejenstam et al. [7] reported that after nearly one year of corrosion testing in liquid lead, 15Cr-Nb AFA alloys exhibited significantly reduced corrosion compared to 15-15Ti austenitic steel, forming an ultrathin oxide layer containing k-Al_2_O_3_ or θ-Al_2_O_3_. Al-containing oxide films [8] comprise Fe/Cr oxides, Cr/Al oxides, and Al_2_O_3_. Beneath the oxide film, a NiAl phase acts as a reservoir for Al diffusion to sustain the growth of the Al_2_O_3_ protective layer, providing long-term self-healing corrosion resistance [9,10]. Studies in 600 °C liquid lead showed that the AFA alloy surface formed an ultrathin oxide layer (~100 nm) of (Al_0.77_Cr_0.23_)_2_O_3_ [11].

The oxidation mechanism of oxygen-controlled LBE is similar to that of metal oxidation in other oxygen-containing environments [12]. The “available space model” can explain the formation of oxidation zones and ion migration on the alloy surface in LBE [13]. This model suggests that Al can reduce oxygen ingress into the material by forming a continuous and dense oxide film, thereby enhancing corrosion resistance. The rate of alumina formation is controlled by the diffusion of oxygen ions through the oxide layer to the substrate and the migration of metal ions to the oxide–liquid interface [14]. The migration of oxygen through the Al_2_O_3_ layer is governed by the concentration of oxygen vacancies [15]. The presence of multivalent cations with valences higher than Al^3+^ reduces the concentration of oxygen vacancies to compensate for the increased positive charge caused by aliovalent cation impurities. This suppresses the growth rate of the alumina film. These findings suggest that the formation of a stable oxide film requires not only a high thermodynamic driving force but also consideration of the oxide layer formation rate. The resistance of materials to lead–bismuth corrosion can be improved kinetically by promoting the formation of Al_2_O_3_ in a competitive relationship between oxidation and dissolution.

Since Ni has a high solubility in liquid lead and Mn, which stabilizes the austenitic structure in a manner similar to Ni, has a solubility in LBE that is about ten times lower than that of Ni, replacing part of the Ni with Mn appears beneficial for mitigating dissolution corrosion. Additionally, Mn substitution for Ni is expected to improve the alloy’s radiation resistance and significantly reduce costs. Mn-substituted AFA alloys exhibit great potential and demand for corrosion resistance in LBE. Studies have shown that in high-temperature air oxidation environments, Mn readily forms external oxide layers [16]. For Mn-containing AFA alloys with approximately 50% reduced Ni content, creep-rupture lifetimes under harsh conditions (1023 K and 100 MPa) range from 7.2 × 10^5^ to 1.8 × 10^6^ s (200–500 h), demonstrating improved performance compared to 347 stainless steel, indicating that mechanical properties remain at a high level after Mn substitution for Ni [17].

However, the substitution of Mn for Ni in AFA steels presents certain challenges. Manganese is an element prone to segregation and exhibits relatively high solubility in Cr_2_O_3_ oxide scales, which can facilitate the transformation of the oxide into a spinel-structured CrMn_1.5_O_4_. In AFA steels, the formation rate of Cr_2_O_3_ oxide scales is faster than that of Al_2_O_3_ oxide scales. Under such conditions, the Cr_2_O_3_ oxide scale may transform into the spinel-structured CrMn_1.5_O_4_ before the formation of the Al_2_O_3_ oxide scale, significantly compromising the oxidation resistance of AFA steels [18,19,20].

Nevertheless, studies have also indicated that the solubility of Mn in Al_2_O_3_ oxide scales is nearly negligible, which does not impair the protective effect of the alumina oxide scale. Additionally, the other properties of austenitic stainless steels with Mn substituting for Ni remain acceptable [21,22,23,24]. This makes it feasible to develop cost-effective, high-performance AFA steels.

A critical issue in developing Mn-containing AFA steels is determining the appropriate Mn content. An increase in Mn content can inhibit the precipitation of the NiAl phase, which is one of the sources of aluminum for the formation of Al_2_O_3_ oxide scales. Consequently, higher Mn content may suppress the formation of alumina oxide scales [25]. Moreover, if Mn-containing AFA steels are to be used in liquid metal environments, electrochemical corrosion cannot be overlooked. Research has shown that Mn can form non-metallic inclusions such as MnS with sulfur, and the potential difference between these inclusions and the matrix can lead to severe electrochemical reactions, degrading the corrosion resistance of the matrix [26,27].

This study prepared various novel Mn-containing AFA alloys with low Cr and Ni content, investigating their corrosion resistance in high-temperature LBE (liquid bismuth–lead eutectic) environments, with the aim of identifying Mn-containing AFA steels suitable for application in high-temperature LBE conditions.

## 2. Experiment

### 2.1. Alloy Preparation

The novel austenitic heat-resistant steel, AFA, was prepared using a vacuum induction melting method. The design composition is shown in Table 1. Several alloy variants were considered, named AFA1-MnY, AFA2-Y, AFA3-Mn, and AFA4, with the primary distinction being the presence or absence of the Mn and Y elements. The raw materials for preparing the AFA alloy consisted of high-purity metal blocks, including Fe, Cr, Ni, Mo, Al, Nb, etc., with purity levels ranging from 99.9% to 99.99%. Prior to melting, the metal blocks were polished with sandpaper or a grinding wheel to remove surface oxides and then cleaned with alcohol to remove any stains or impurities.

The alloy ingots were prepared using a vacuum induction melting furnace (ZGL-0.01) with the following parameters: 4000 Hz, 50 kW, and a 10 kg capacity. The specific melting process is as follows: First, the charge materials are loaded into the melting furnace, which is then evacuated to a vacuum level of 3 Pa. Subsequently, argon gas is introduced to a pressure of 0.05 MPa, and electrical power is supplied to initiate the melting process. After the alloy is completely molten, it is refined for 3–5 min before being cast into molds (spheroidal graphite iron molds). The final alloy ingot obtained is shown in Figure 1.

To reduce internal structural defects in the ingots, the obtained alloy ingots were heated to 1200 °C in the furnace, maintained for 1–2 h, and then forged. The actual forging temperature was approximately 1180–1190 °C, with a final forging temperature of 1100 °C. The forging ratio was no less than 3:1. The ingots were forged into round bar samples using an air hammer, with a diameter of 60–70 mm and an unlimited length. The forged alloy samples are shown in Figure 2. The chemical composition of the forged samples was analyzed, and the results are shown in Table 2. Comparison with Table 1 reveals that some element loss occurred, but the material was essentially close to the expected design composition.

### 2.2. Microstructural Analysis

#### 2.2.1. XRD

The XRD phase diagrams of the four types of AFA alloy show a high degree of consistency, as shown in Figure 3. The diffraction peaks are the same, indicating that the matrix phase of the alloys is identical. All exhibit distinct austenitic peaks, with the main peak at 43° corresponding to the (111) plane of austenite, a peak at 50° corresponding to the (200) plane of austenite, and a peak at 75° corresponding to the (220) plane of austenite, all of which are typical of the γ face-centered cubic structure.

#### 2.2.2. EBSD

Electron Backscattered Diffraction (EBSD) was utilized to further analyze the microstructure of the crystals. After polishing and smoothing the samples, they were electrolytically polished using a solution of 10% HClO_4_ + 90% C_2_H_5_OH (by volume) and observed under SEM + EBSD to examine the grain structures of the sample surfaces, as shown in Figure 4.

From the phase determination in Figure 4a–d, it is clear that all alloys consist of a single austenite face-centered cubic structure, consistent with the theoretical design calculations and metallographic results, achieving the design goal. The graphs in Figure 4e,f indicate that, apart from AFA2-Mn, the grain size distribution of the forged AFA alloys is non-uniform, with large grains interspersed with fine grains, and the overall grain size of AFA2-Mn is smaller than that of the other alloys. The grain size distribution is illustrated in Figure 5, with the corresponding statistical data presented in Table 3.

The grain size of AFA1-MnY varies significantly, with generally larger grains. The grain size of AFA2-Y is more uniform compared to the other alloys, with most grains distributed in the range of 13 μm to 125 μm, and the proportions of grains of each size are relatively similar. In the AFA3-Mn alloy, about 22.3% of the grains are smaller than 40 μm, while the distribution of grains in the range of 40–307 μm is quite uniform. In the AFA4 alloy, the proportion of small grains is the largest. It is evident that larger grains exist in the alloy structure with the addition of Mn, leading to a more pronounced occurrence of a bimodal distribution of grain sizes. The presence of Mn promotes grain growth, and severe segregation of Mn in steel can lead to the formation of mixed grains. At high temperatures, the segregation of alloying elements can cause compositional non-uniformity, with varying degrees of austenitization in different regions. The grains in the areas that transform to austenite first continue to grow, while those in areas that have not fully transformed remain small. If austenitization is prematurely terminated and rapid cooling then takes place, a coexistence of large and small grains can occur. In the Mn-containing alloys shown in Figure 4, the grain sizes are not uniform; the differences in grain size among the four alloys relate to the intrinsic heterogeneous size of the crystals and the sizes of the EBSD-selected regions. The cooling rate during the solidification process also influences this, as a fast cooling rate results in equiaxed grain boundaries forming at lower temperatures, leading to finer grains.

Fine grains can enhance the yield strength, fatigue strength, plasticity, and impact toughness of metallic materials while reducing the brittle transition temperature, thereby playing a role in grain refinement strengthening [28]. Smaller metallic grains lead to a greater total area occupied by grain boundaries and more dislocation obstacles, requiring more coordination among grains with different orientations, which increases the resistance to plastic deformation of the metal. Conversely, prolonged holding times at high temperatures can lead to grain coarsening, where large grains diminish the mechanical properties of the material. The formation of mixed grains can cause instability in the material’s mechanical performance, highlighting the need for refinement and control in the preparation process, alongside adjustments via subsequent heat treatment.

Figure 6 and Figure 7 present the microstructure and grain boundary angle distribution of the four types of AFA alloy. The EBSD images show a higher presence of low-angle grain boundaries, indicated by blue lines. In AFA2-Y, besides low-angle grain boundaries, there is also a significant proportion of high-angle grain boundaries, with 26.6% of boundaries being in the 2–5° range and 67.8% being greater than 15°, with more uniform distribution of intermediate-angle boundaries correlating with the grain boundary size distribution. The other three AFA alloys exhibited relatively consistent test results with a predominance of low-angle grain boundaries. In the AFA1-MnY alloy, the proportion of 2–5° boundaries is 71.3%, while those over 15° constitute 9.4%; in the AFA3-Mn alloy, the 2–5° boundaries are 61.5%, and those over 15° are 23.5%; for the AFA4 alloy, 2–5° boundaries account for 56.9%, and those over 15° account for 26.0%. Low-angle grain boundaries have relatively low energy and typically exhibit elongated oriented arrangements. Due to the ease of element diffusion along these boundaries, they may be advantageous for corrosion resistance, while high-angle grain boundaries are associated with the recrystallization of grains.

### 2.3. Corrosion Experiments

#### 2.3.1. Sample Processing

The materials used in the experiment include the AFA steel prepared in the laboratory, as described in the previous section. The corrosion samples were obtained by machining bulk AFA steel samples and then finely processing them to create smooth surfaces on both the top and side faces for corrosion testing. The samples were cylindrical in shape, with dimensions of Ø8 mm × 15 mm. During the experiment, both the top and bottom surfaces of the samples were processed with through holes to facilitate their connection.

#### 2.3.2. Corrosion Testing Method

The objective of the experiment was to test the corrosion behavior of the alloy under 10^−6^ wt.% oxygen control conditions and to assess the effect of Mn addition on the corrosion resistance of AFA steel.

The corrosion tests were conducted in a static corrosion device. The main testing equipment was a stainless-steel corrosion vessel, where the samples were suspended on a sample rack and placed into the corrosion chamber via an inlet port. The sample rack was positioned in the temperature-controlled region. Two ports were used to introduce thermocouples for temperature control, with the thermocouples placed in slender alumina tubes. One or two ports were used for placing electrochemical oxygen sensors (depending on the number of test samples and the volume of the container) to control the electrode potential during the experiment. The oxygen content inside the chamber was adjusted by a mixture of hydrogen and oxygen gases, and the oxygen sensor provided real-time output. The zirconia solid electrolyte on the oxygen sensor came into contact with the liquid metal. The internal structure of the corrosion device is shown in Figure 8, and the oxygen sensor is shown in Figure 9. Samples, oxygen sensors, and thermocouples were immersed in liquid metal. The distance from the bottom of the lowest sample to the bottom of the reaction vessel, as well as the distance from the top of the highest sample to the liquid metal surface, was recorded. The electrode voltages were monitored during the test.

The equipment satisfied the following conditions during the experiment: (a) All samples being tested simultaneously were placed in the temperature-controlled zone of the testing device, where local temperature variations did not exceed +/− 1(2) °C. The average temperature in this zone did not differ from the nominal temperature by more than +/− 3 °C. (b) The liquid metal should be free from Ni contamination, especially in tests involving aluminum-containing steel (to prevent the formation of Ni-Al intermetallic compounds that could affect the experimental results).

#### 2.3.3. Post-Corrosion Sample Treatment

After completing the corrosion tests, the samples were removed from the LBE corrosion vessel when the temperature had reduced to around 200 °C. The samples were naturally cooled to room temperature in an argon environment. A precision cutting machine was used to cut the samples axially into two segments. One segment was cold-mounted and polished to observe the microstructure. The other segment was chemically cleaned using a mixture of heated glycerol (below 150 °C) and a CH_3_COOH-H_2_O_2_-C_2_H_5_OH (1:1:1) solution at room temperature to remove the adhering liquid metal. The surface corrosion morphology of the cleaned samples was then observed, and their oxidation products were analyzed. By combining the data and results from the analysis process, quantitative data for each sample under different corrosion conditions were obtained, the corrosion types were identified, and their corrosion resistance was compared.

## 3. Results and Discussion

### 3.1. Mass Changes Before and After Corrosion

The weight changes of the four alloy samples before and after corrosion are presented in Table 4. Each sample includes three replicates, and the balance used for weighing has an accuracy of up to 10 μg. As seen in Table 4, the AFA3-C-Mn sample exhibited a gain in weight, while the other samples, namely AFA1-C-MnY, AFA2-C-Y, and AFA4-C, experienced weight loss after corrosion. The weight gain after corrosion is attributed to the formation of oxide products due to oxidation reactions. When dissolution exceeds oxidation, the overall weight decreases; conversely, when oxidation predominates, the weight increases.

Based on the data presented in Table 4, the mean values and standard deviations of weight changes were calculated. The results indicate that the weight change ratio of AFA2-C-Y was the highest, at −0.64% ± 0.13%. In contrast, the weight changes of AFA4-C and AFA1-C-MnY were relatively minor, with values of −0.052% ± 0.0086% and −0.052% ± 0.012%, respectively. The smallest weight change was observed for AFA3-C-Mn, with a value of 0.0075% ± 0.002%, suggesting that the mass remained virtually unchanged before and after the experiment. Notably, AFA1-C-MnY and AFA3-C-Mn, which exhibited minimal weight changes, are both Mn-containing alloys. This observation suggests that the addition of Mn under controlled oxygen conditions plays a positive role in enhancing the corrosion resistance of these alloys.

### 3.2. Influence of Mn on the Microstructure of Corrosion Cross-Sections

#### 3.2.1. Mn-Free Alloys

The scanning electron microscopy (SEM) analysis of the corrosion region in AFA2-C-Y is presented in Figure 10 and Table 5. In the entire cross-section, no Pb or Bi was detected (0 wt.%). Elemental mapping revealed a significant oxygen (O) signal, indicating that lead–bismuth eutectic (LBE) penetration into the cross-section did not occur. The EDS line scan in Figure 10a shows that on both sides of the original boundary, the oxide layer on the alloy surface consists of an outer oxide layer and an inner oxide layer. The outer oxide layer contains only Fe and Cr, while the inner oxide layer is more complex, containing Al, Cr, and Fe. Elemental mapping in Figure 10 shows that the inner oxide layer exhibits significant Fe depletion, with O diffusing into this region. The corrosion mechanism observed at this stage is similar to that reported by Wang et al. [28,29] for the corrosion of 316 austenitic stainless steel, with the formation of a bilayer oxide structure. This corrosion mechanism can be generally explained by the model proposed by Bischoff et al. [30,31,32,33].

At the onset of oxidation corrosion, oxygen in the liquid lead–bismuth eutectic (LBE) reacts with the surface of AFA steel to form an extremely thin oxide film (Fe_2_O_3_). Once the oxide film is formed, it isolates the metallic substrate from the oxygen. The reactants (oxygen ions and cations) then diffuse through the oxide film, leading to the growth of the oxide in two directions (outward and inward). The outer oxide layer is formed by the outward diffusion of Fe cations, while the inner oxide layer is formed by the inward diffusion of oxygen ions (O) that oxidize the underlying substrate.

In the initial stage of corrosion, oxygen in LBE is absorbed by the surface of the substrate, generating iron vacancies. These vacancies migrate inward with electron holes, forcing Fe to move outward. Over time, Fe diffuses outward to form an Fe_3_O_4_ outer oxide layer, while O diffuses inward to form an inner oxide layer that is enriched in Al and Cr and depleted in Fe.

From the EDS point analysis in Table 5, the elemental composition at point P4, which is in the bulk alloy, remained largely unchanged. Point P3 represents the outward-growing oxide product, with an O content of 29.93 wt.% and an Fe content of 66.02 wt.%. Cr and Al contents were minimal, indicating that the oxide product was magnetite (Fe_3_O_4_) [33]. The EDS line scan showed that the trends in Fe and O were consistent, suggesting a relatively simple oxide structure. Beneath the initial interface, a mixed oxide layer formed from (Fe,Cr)_x_O_y_ and (Fe,Al)_x_O_y_. This inner oxide layer was composed of oxidation products containing Cr and Al [34].

The Ni content from points P1 to P2 and P5 decreased as follows: 23.93 wt.% → 16.17 wt.% → 13.43 wt.%. This indicates that from the base alloy → base alloy/inner oxide layer boundary → inner oxide layer/outer oxide layer boundary, the Ni content first increased and then gradually decreased. This trend is related to significant Fe diffusion outward to form Fe3O4, leaving vacancies within the matrix. Ni is then diffused to fill these vacancies, resulting in higher Ni content at the base alloy/inner oxide layer boundary. However, Ni content decreased near the inner/outer oxide layer boundary due to outward diffusion and dissolution into the LBE. Meanwhile, the vacancies created by Fe diffusion provided pathways for O diffusion into the matrix, which reacted with Cr and Al to form the inner oxide layer. The inward-diffusing Ni was hindered from further outward diffusion in this region, leading to relatively high Ni content at the boundary.

The corrosion region of AFA4-C was analyzed via SEM, as shown in Figure 11. Compared to AFA2-C-Y, AFA4-C exhibited reduced surface corrosion, with localized corrosion observed in some regions. This could be attributed to the lower Ni content in AFA4-C. Elemental mapping revealed that an Fe-containing outward-growing oxide layer formed, beneath which was an inner oxide product. In the localized corrosion region, the O signal was prominent, and the oxide layer was divided into an outer and an inner oxide layer by Cr distribution. Specifically, the outer oxide layer primarily consisted of Fe and O, forming a magnetite (Fe_3_O_4_) structure. The inner oxide layer showed significant dissolution of Ni and Fe, with Cr and Al oxides forming as primary products.

#### 3.2.2. Mn-Containing Alloys

The corrosion region of AFA1-C-MnY was analyzed using SEM, as shown in Figure 12 and Table 6. A relatively uniform micrometer-thick oxide layer formed on the surface, with localized oxidation occurring in some regions (Figure 12b). The penetration of Pb and Bi into the oxide layer was minimal. The EDS line scan (Figure 12c) revealed that, similar to the results obtained without Mn, the oxide layer was divided into two sublayers: the outer layer consisted primarily of Fe and O and was identified as magnetite (Fe_3_O_4_); the inner layer was more complex, containing Fe, Cr, Mn, and Al, forming a mixed oxide product.

EDS point analysis (Table 6) showed that at point P2, within the inner oxide layer, the Cr content was 17.66 wt.% and the Mn content was 6.18 wt.%. Closer to the bulk alloy, the Cr and Mn contents decreased slightly, while the Al content increased. This indicates that Mn-containing oxides, similar to Cr-containing oxides, tend to form on the outer side of Al-containing oxides. The inner oxide layer likely consisted of a mixture of (Fe,Cr)_x_O_y_, (Fe,Al)_x_O_y_, and MnO. Al-containing oxides were closer to the alloy substrate, while MnO was nearer the outer side. The oxygen content decreased toward the substrate, indicating that the oxide layer effectively hindered inward O diffusion.

The outer and inner oxide layers of AFA1-C-MnY showed slightly higher Ni content than the bulk alloy, but no Ni was detected within either oxide layer. Under oxygen-controlled conditions (10^−6^ wt.% oxygen), Ni was not oxidized. After the initial oxide layer formed, Ni migration from the substrate to the oxide layer boundary was hindered, leading to a trend in Ni content that first increased and then decreased, similar to Mn-free AFA alloys.

The SEM analysis results of the corrosion region in AFA3-C-Mn are shown in Figure 13 and Table 7. The entire region of the AFA3-C-Mn sample has a thinner oxide layer compared to AFA1-C-MnY, with no noticeable localized corrosion areas. The oxide layer is continuous and dense. From the line scan, it is evident that in the regions with thinner oxide layers, the overall thickness is less than 500 nm. The Energy Dispersive X-ray Spectroscopy (EDS) line scan in panel (b) reveals that the oxide layer under SEM does not exhibit distinct layering, and no significant outward Fe_3_O_4_ formation is observed. Instead, a clear oxide layer containing Al and Mn has formed, which is notably different from the oxide layers formed in other alloys.

In the EDS point analysis shown in Table 7, point P3 corresponds to the uncorroded base material, where the Bi and O signals are impurity signals. At point P1, the Cr and Mn content is relatively high, suggesting the presence of (Fe,Cr)_3_O_4_ and MnO products. Point P2, which contains a substantial amount of Al, is likely to be the (Fe,Al)_3_O_4_ oxidation product. The results indicate that the nanometer-scale oxide layer is not a single structure but has a finer layered structure. Point P1 is located on the outer part of the oxide layer, indicating that the Mn-containing oxides form the outer layer, while the inner region of the oxide layer is rich in Al.

The oxide layer structure of AFA3-C-Mn differs from that of other alloys. The addition of Mn inhibits the formation of Fe_3_O_4_ oxidation products, promoting the formation of a continuous Al-containing oxide layer, which enhances the alloy’s corrosion resistance. The migration of Ni from the substrate to the oxide layer boundary is hindered, causing Ni to accumulate beneath the oxide layer. This results in a trend of increasing Ni content from the base material to the outer surface, followed by a decrease. Unlike the AFA alloy without Mn, the Ni content does not increase further, which is related to the absence of Fe_3_O_4_.

#### 3.2.3. Structure of the Protective Oxide Film

The AFA3-C-Mn sample, which has a thin continuous oxide layer, was thinned by Focused Ion Beam (FIB) as shown in Figure 14. The thinning region, about 6 μm × 6 μm in size, was thinned to a thickness of approximately 70 nm. Transmission Electron Microscopy (TEM) was used to analyze the microstructure using diffraction and high-resolution techniques, along with accompanying EDS for line scans and surface distribution analysis.

In the Scanning Transmission Electron Microscopy (STEM) mode, the thinned region (shown in panel (a)) exhibits few pores, and the sample thickness is relatively uniform, indicating successful FIB thinning. Upon magnification of the thinned region (panel (b)), the right side corresponds to the alloy’s outer surface, which is in contact with LBE. This part was in contact with the embedding material during cold mounting. The left side represents the uncorroded steel substrate, which has a more uniform microstructure. Panel (c) highlights the continuous oxide layer formed on the alloy’s surface. TEM diffraction analysis of the middle point of the continuous oxide layer is shown in Figure 15. The diffraction pattern of the oxide layer is characterized by scattered diffraction spots but also a ring-like polycrystalline diffraction pattern, indicating a polycrystalline structure. The diffraction spots form concentric rings, typical of a polycrystalline structure with numerous crystals oriented in different directions.

A diffraction analysis was conducted on the region below the oxide layer, which exhibits regular and orderly morphology, as shown in Figure 16. The diffraction pattern indicates a single-crystal structure. After diffraction calibration and comparison with PDF cards, the structure was identified as face-centered cubic (FCC) austenitic phase (Cubic, Fm-3m (225), γ-Fe [0 2]), PDF#52-0512. This region corresponds to the uncorroded substrate, confirming the protective role of the oxide layer.

STEM analysis was conducted to observe the elemental distribution of the thinned region, as shown in Figure 17. The oxide film thickness is less than 1 μm, and the outer layer of the oxide is irregular, with no Fe_3_O_4_ product growing outward. The middle layer is continuous, dense, and more regular, primarily consisting of Cr and Mn oxidation products. Based on the polycrystalline structure, it is hypothesized that the oxide product may be (Cr,Mn)_x_O_y_. The continuous oxide layer has a thickness of 200–300 nm, while the inner part of the oxide layer shows internal oxidation of Al, with a thickness of about 600 nm. In the inner part of the continuous oxide layer, a small amount of Al_2_O_3_ is visible, with a thickness of less than 50 nm. This could be due to the substantial thickness of the Cr/Mn oxide layer, which effectively inhibits the diffusion of oxygen, resulting in low oxygen content penetrating to the inner part and minimal formation of Al_2_O_3_. Long-term corrosion experiments are required to confirm whether a stable continuous Al_2_O_3_ oxide film will form in this region.

At the bottom of the oxide layer, high-resolution observations were made near the oxide layer, as shown in Figure 18. No coarse precipitates were observed; instead, small nanometer-sized precipitates are still present, indicating that the precipitates are stable during corrosion and suggesting the effective corrosion resistance of the oxide layer.

STEM-EDS analysis of the corrosion region is shown in Figure 19 and Table 8. The contrast of the oxide layer in STEM mode is clearly visible, with a continuous gray oxide layer gradually separating the outer surface and the substrate. Below the continuous oxide layer, a black, elongated line of oxidation products is observed, with irregular morphologies at both ends of the continuous oxide layer. The EDS line scan reveals that oxygen content is highest in the continuous oxide layer region. The black, elongated products contain Al_2_O_3_ but do not form a continuous Al_2_O_3_ oxide film.

EDS line and point scans were conducted from the outer surface to the substrate. Point 1 contains almost only Bi, corresponding to the solidified lead–bismuth adhering to the surface. At point 5, the O content is 32.15 wt.%, Cr is 32.21 wt.%, Mn is 13.55 wt.%, and Pb is 9.55 wt.%, indicating a mixture of Cr/Mn oxidation products and Pb. Some Pb has penetrated into this oxide product. At point 7, the O content is 85.51 wt.%, with trace amounts of other elements, likely representing voids in the oxide product. Point 2, with O content of 37.08 wt.%, Cr of 41.92 wt.%, and Mn of 16.96 wt.%, contains no Pb or Bi, suggesting the continuous oxide layer effectively resists LBE penetration, forming Cr/Mn oxidation products. Point 8 corresponds to undissolved and unoxidized Fe, which further reflects the ability of the oxide layer to impede oxygen and alloy element interdiffusion. Points 9 and 10, with similar compositions, correspond to Cr/Mn oxidation products, and points 3 and 4 correspond to the undissolved base material, similar to point 8. Point 6, with O content of 44.31 wt.%, Cr of 14.47 wt.%, Mn of 28.09 wt.%, and Al of 9.56 wt.%, is identified as a mixed product of Cr/Mn oxidation and internal Al oxidation.

Combining the previous analysis, it is clear that AFA3-C-Mn demonstrates strong corrosion resistance to lead–bismuth eutectic (LBE). It forms a continuous Cr/Mn oxide layer, with a very thin discontinuous Al_2_O_3_ layer beneath it. The internal oxidation region is less than 1 μm thick. The addition of Mn alters the type of oxide product formed, shifting from Fe_3_O_4_ to Cr/Mn oxide products.

Overall, the corrosion resistance of various alloys follows the following order: AFA3-C-Mn > AFA1-C-MnY > AFA4-C > AFA2-C-Y. AFA alloys form micron- and nanometer-sized Al-containing oxide films, which have complex multi-layer structures [8].

## 4. Conclusions

The corrosion performance of four different AFA steel alloys at 550 °C in 10^−6^ wt.% oxygen-controlled LBE was analyzed. The corrosion behavior after 1000 h of exposure was quantified, and the severity of corrosion for each sample was compared. The morphologies of the oxides formed on the surfaces of the alloys were analyzed to determine the effects of different alloy compositions on corrosion resistance. The conclusions are as follows:Under the condition of 10^−6^ wt.%, Mn has a more significant beneficial effect on the corrosion resistance of AFA alloys. The order of weight change for the alloys is as follows: AFA2-C-Y > AFA4-C > AFA1-C-MnY > AFA3-C-Mn. Under 10^−6^ wt.%, the time for dissolution-induced weight loss is longer. The weight loss of AFA1-C-MnY, AFA2-C-Y, and AFA4-C alloys increases after corrosion, whereas the weight of AFA3-C-Mn increases slightly due to the early formation of a dense oxide film, which effectively hinders the dissolution of alloy elements.No obvious LBE infiltration was observed in any of the AFA alloys, and the degree of oxidation was reduced. The order of corrosion resistance is as follows: AFA3-C-Mn > AFA1-C-MnY > AFA4-C > AFA2-C-Y. Under 10^−6^ wt.% oxygen-controlled conditions, the surface of the alloys does not directly form distinct Fe_3_O_4_ magnetite via reaction between Fe and O.The corrosion of AFA4-C is less severe than that of AFA2-C-Y, with some localized corrosion appearing in certain regions. This is related to the lower Ni content in AFA4-C.AFA1-C-MnY exhibited slight localized oxidation, with the outer layer mainly composed of Fe_3_O_4_ magnetite. The inner layer had a complex structure, with mixed oxidation products such as (Fe,Cr)_x_O_y_, (Fe,Al)_x_O_y_, and MnO. The Al-containing oxide product was closer to the substrate, while MnO was closer to the surface. No obvious oxide layer was observed in other areas. The entire surface of AFA3-C-Mn showed no significant corrosion. It exhibited the strongest corrosion resistance, forming a continuous Cr/Mn oxide layer. Beneath this layer was an extremely thin, discontinuous Al_2_O_3_ oxide layer, with small internal oxidation areas (<1 μm) in the alloy. The addition of Mn altered the type of oxide products formed on the surface, shifting from Fe_3_O_4_ to Cr/Mn oxide products.

## Figures and Tables

**Figure 1 materials-18-01328-f001:**
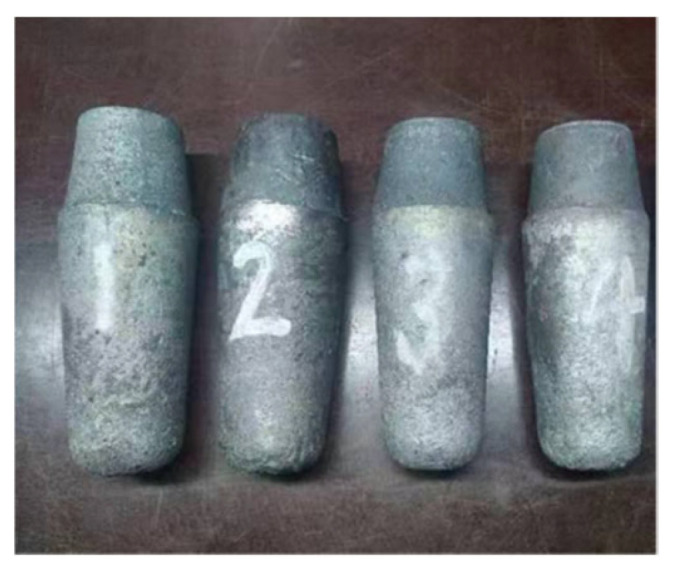
Alloy ingots from melting.

**Figure 2 materials-18-01328-f002:**
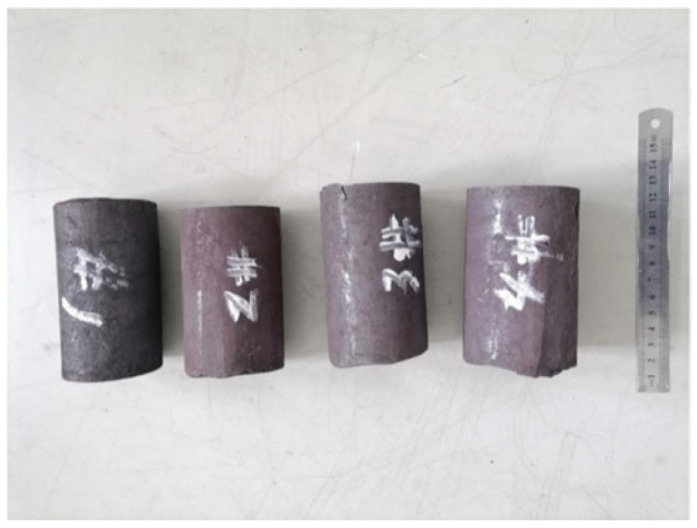
Forged alloy samples.

**Figure 3 materials-18-01328-f003:**
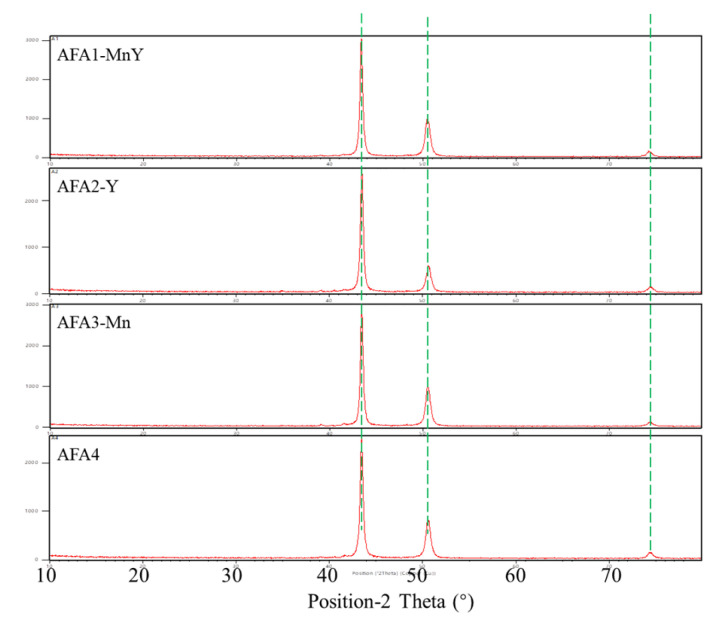
XRD phase diagram of AFA alloys.

**Figure 4 materials-18-01328-f004:**
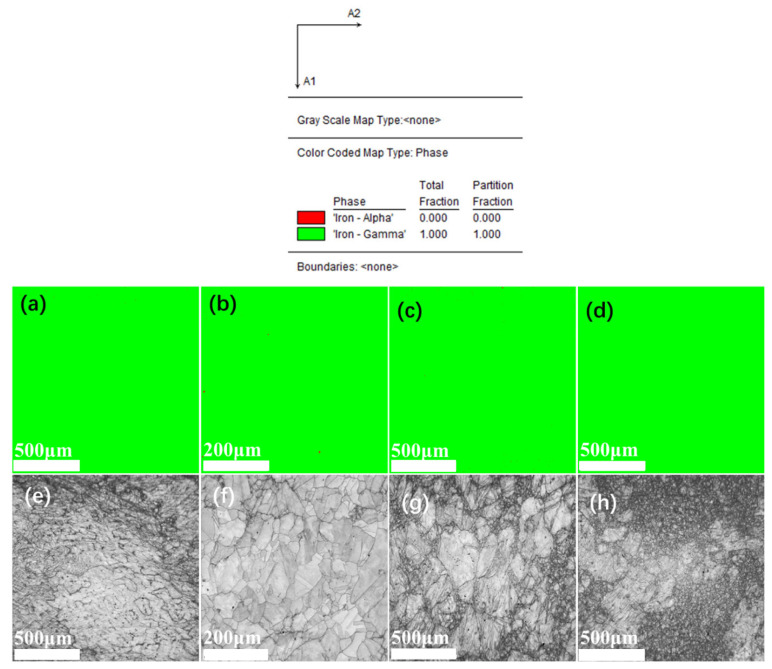
Phase structure and grain morphology size distribution of AFA alloys: (**a**) the phase structure of the AFA1-MnY alloy; (**b**) the phase structure of the AFA2-Y alloy; (**c**) the phase structure of the AFA3-Mn alloy; (**d**) the phase structure of the AFA4 alloy; (**e**) the grain morphology of the AFA1-MnY alloy; (**f**) the grain morphology of the AFA2-Y alloy; (**g**) the grain morphology of the AFA3-Mn alloy; (**h**) the grain morphology of the AFA4 alloy.

**Figure 5 materials-18-01328-f005:**
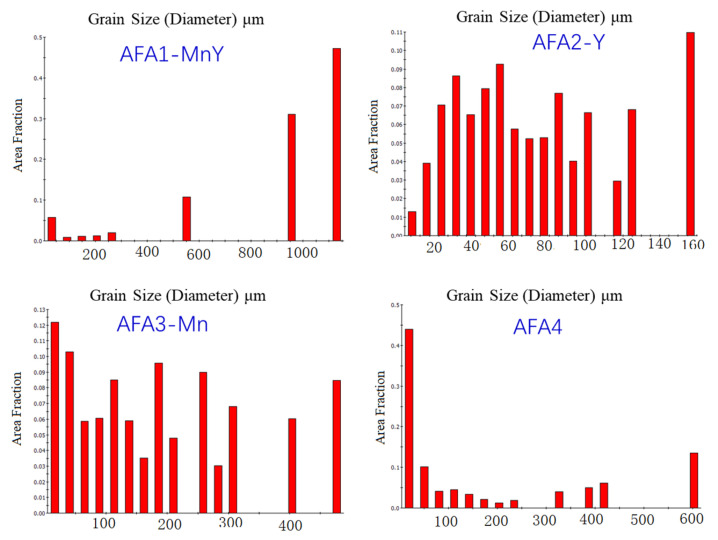
Grain size distribution chart of the alloys.

**Figure 6 materials-18-01328-f006:**
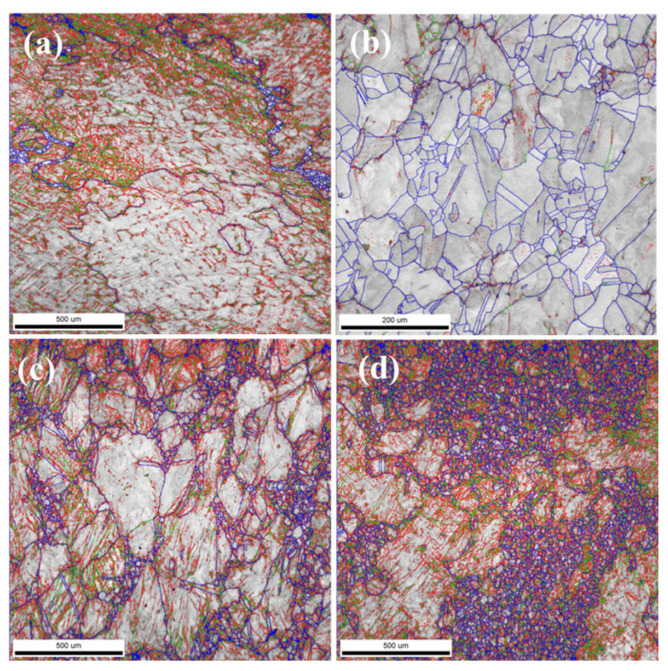
Microstructure of AFA alloys: (**a**) the metallographic photograph of AFA1-MnYalloys, (**b**) the metallographic photograph of AFA2-Yalloys, (**c**) the metallographic photograph of AFA3-Mnalloys, (**d**) the metallographic photograph of for AFA4alloys.

**Figure 7 materials-18-01328-f007:**
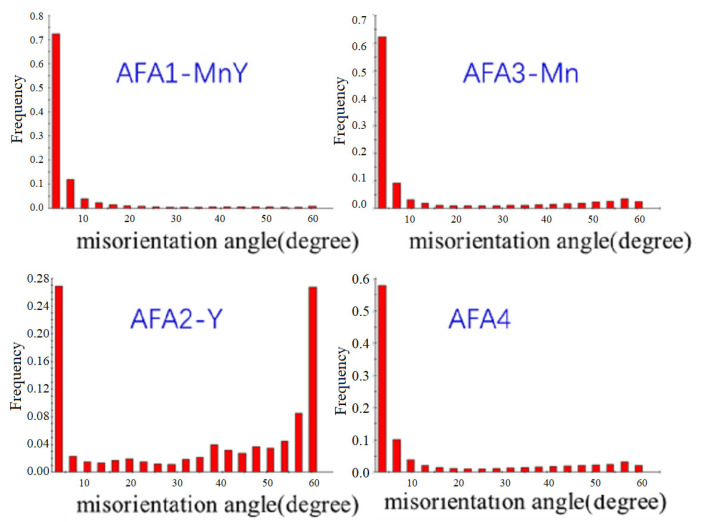
Grain boundary angle distribution chart of AFA alloys.

**Figure 8 materials-18-01328-f008:**
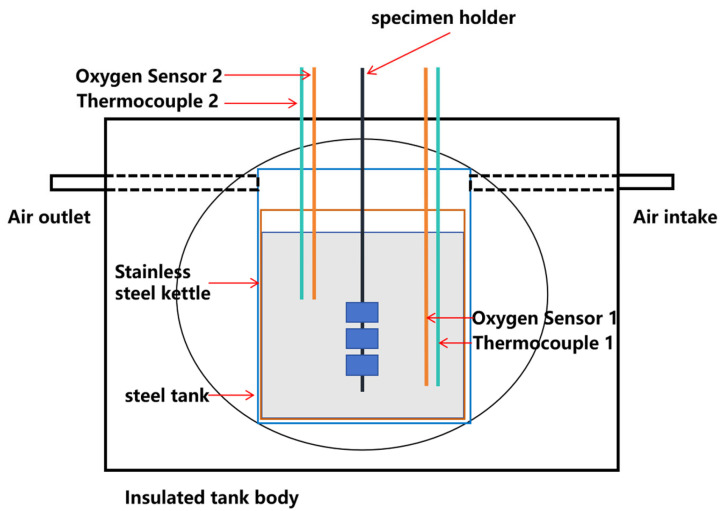
Schematic of the corrosion device.

**Figure 9 materials-18-01328-f009:**
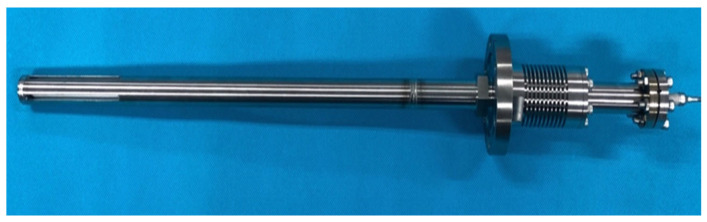
Image of the oxygen sensor.

**Figure 10 materials-18-01328-f010:**
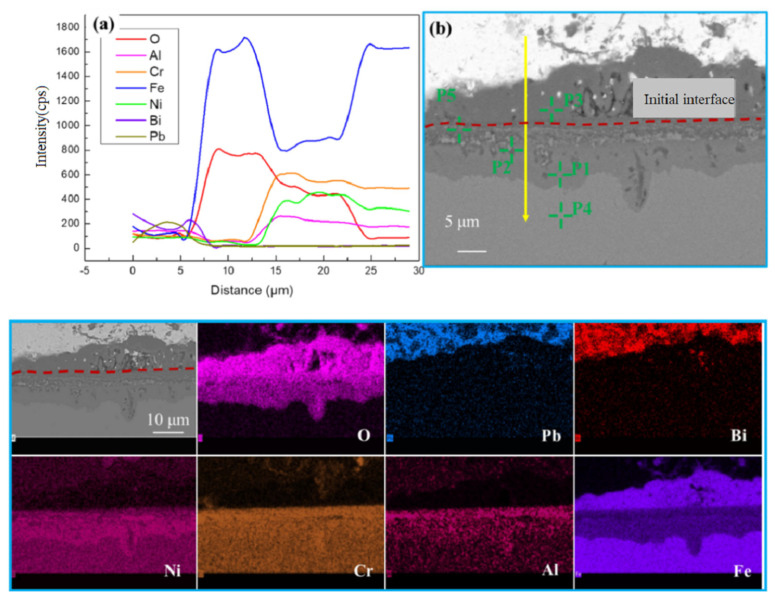
Elemental distribution in the corrosion products of AFA2-C-Y: (**a**) Distribution of elements with depth; (**b**) Scanning electron microscope (SEM) images of the AFA2-C-Y alloy.

**Figure 11 materials-18-01328-f011:**
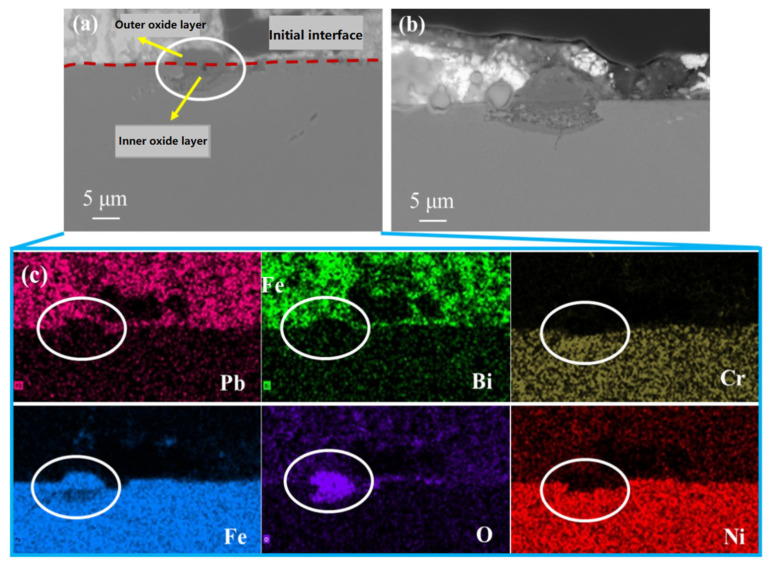
Elemental distribution in the corrosion products of AFA4-C: (**a**) SEM images of the oxide film interface of the AFA4-C alloy; (**b**) the local magnified image of the circled area in subfigure (**a**); (**c**) the elemental mapping results of the circled area in subfigure (**a**).

**Figure 12 materials-18-01328-f012:**
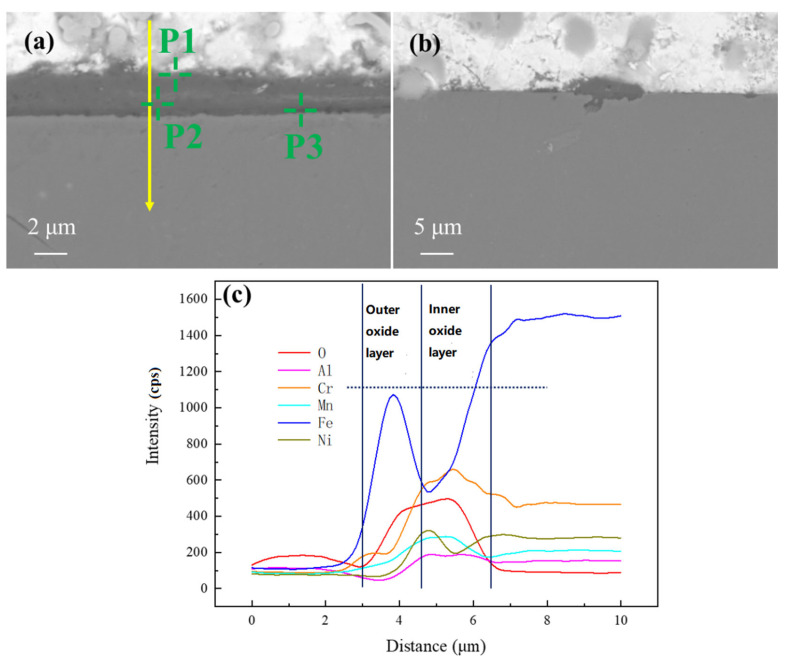
Elemental distribution in the corrosion products of AFA1-C-MnY: (**a**) SEM images of the oxide layer of the AFA1-C-MnY alloy; (**b**) SEM images of the local oxide on the surface of the AFA1-C-MnY alloy. (**c**) EDS line scan results for the area shown in subfigure (**a**).

**Figure 13 materials-18-01328-f013:**
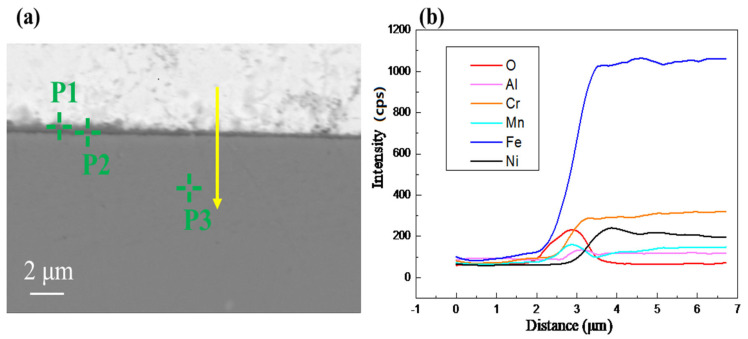
Elemental distribution of corrosion products on AFA3-C-Mn surface: (**a**) SEM images of the oxide layer of the AFA1-C-MnY alloy; (**b**) EDS line scan results for the area shown in subfigure (**a**).

**Figure 14 materials-18-01328-f014:**
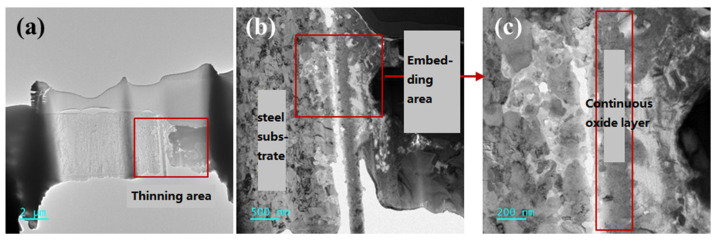
(**a**) TEM bright field image of FIB-sectioned corrosion region of AFA3-C-Mn; (**b**) The local magnified view of the marked area in subfigure (**a**); (**c**) The local magnified view of the marked area in subfigure (**b**).

**Figure 15 materials-18-01328-f015:**
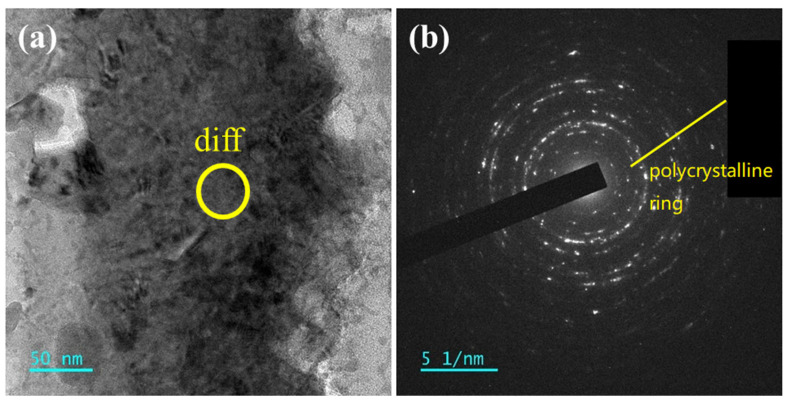
(**a**) TEM images of the continuous oxide layer in the cross-section of the AFA3-C-Mn alloy; (**b**) TEM diffraction pattern of continuous oxide layer on AFA3-C-Mn cross-section.

**Figure 16 materials-18-01328-f016:**
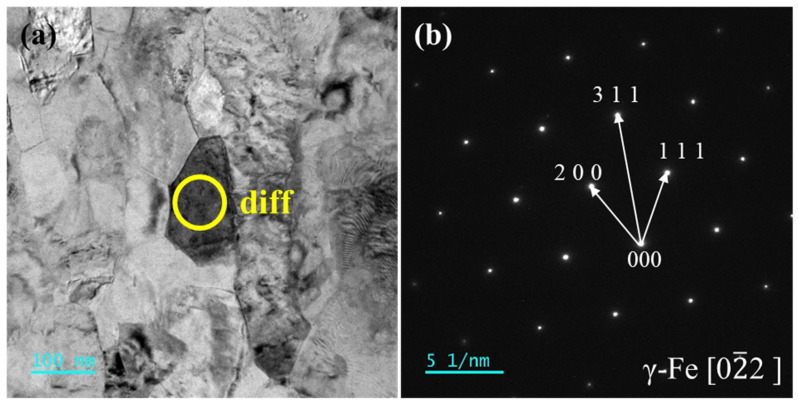
(**a**) TEM images of the substrate region beneath the continuous oxide layer in the cross-section of the AFA3-C-Mn alloy; (**b**) Diffraction pattern of substrate region beneath continuous oxide layer on AFA3-C-Mn cross-section.

**Figure 17 materials-18-01328-f017:**
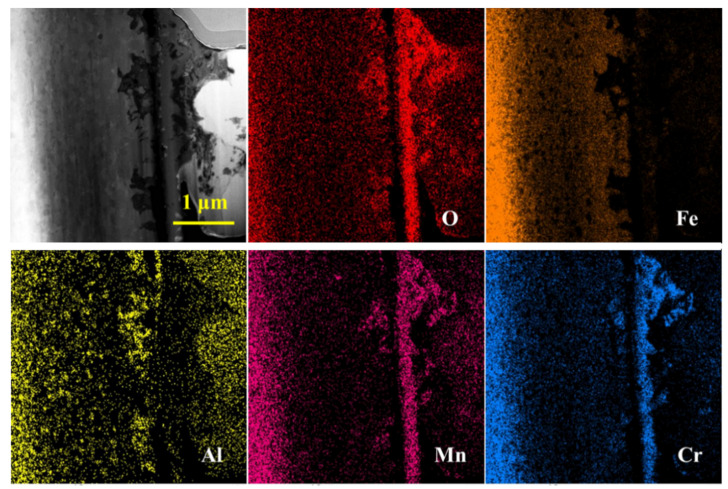
STEM surface element distribution of thinned region in AFA3-C-Mn.

**Figure 18 materials-18-01328-f018:**
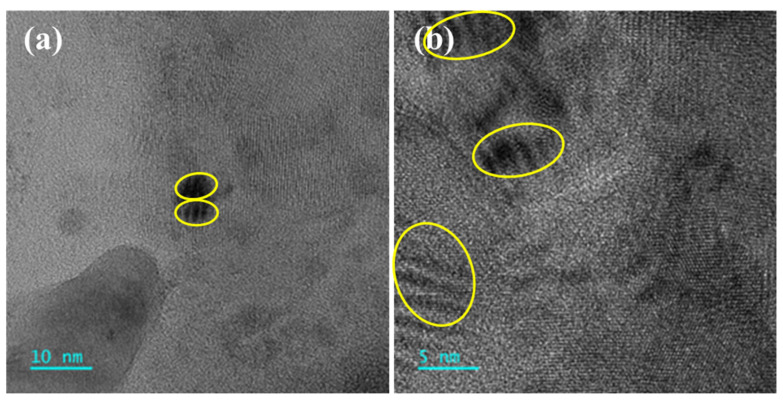
High-resolution images of fine particles beneath oxide layer of AFA3-C-Mn: (**a**,**b**) are TEM images of precipitates from two different regions, with the circled areas in the figures indicating the nanoscale precipitates.

**Figure 19 materials-18-01328-f019:**
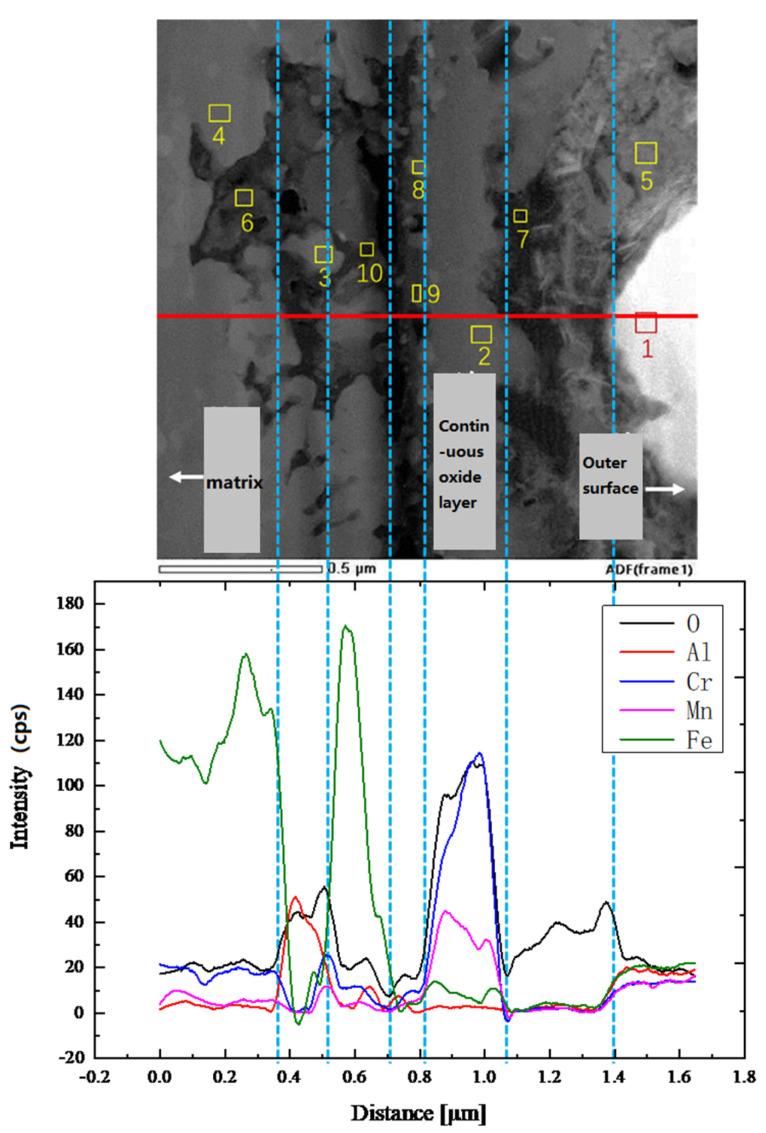
STEM-EDS analysis of oxidation region of AFA3-C-Mn: The points labeled 1 to 10 in the figure represent the signal points for the 10 EDS spot scans, corresponding to the 10 EDS spot scan results listed in Table 8.

**Table 1 materials-18-01328-t001:** AFA steel composition design (wt.%).

Elements	Ni	Cr	Al	Y	Mo	Nb	Mn	C	Fe
AFA1	16	12	3	0.1	2	1	4	0.15	balance
AFA2	18	12	3	0.1	2	1	-	0.15	balance
AFA3	16	12	3	-	2	-	4	-	balance
AFA4	16	12	3	-	2	1	-	0.15	balance

**Table 2 materials-18-01328-t002:** Actual chemical composition of AFA (wt.%).

Elements	Ni	Cr	Al	Y	Mo	Nb	Mn	C	Fe
AFA1	15.30	11.7	2.69	<0.1	2.13	0.98	3.84	0.16	balance
AFA2	16.71	11.5	2.58	<0.1	1.94	0.92	-	0.11	balance
AFA3	15.26	11.6	2.63	-	1.97	-	3.77	-	balance
AFA4	14.76	11.9	2.60	-	2.0	0.88	-	0. 10	balance

**Table 3 materials-18-01328-t003:** Statistical data on grain sizes of the alloys.

Alloy	Maximum Grain Size (μm)	Maximum Grain Size Ratio (%)	Minimum Grain Size (μm)	Minimum Grain Size Ratio (%)
AFA1	>950	78.3	32.9	5.7
AFA2	156.7	11	5.6	1.3
AFA3	477.2	8.5	16.1	12.2
AFA4	602.5	13.5	19.3	43.9

**Table 4 materials-18-01328-t004:** Mass changes of the four alloys before and after corrosion.

	AFA1			AFA2			AFA3			AFA4		
Mass before corrosion (g)	6.1922	6.2652	6.2310	6.3100	6.2613	6.2414	6.2498	6.2525	6.1977	6.3625	6.3009	6.3111
Mass after corrosion (g)	6.1890	6.2625	6.2270	6.2692	6.2311	6.1918	6.2501	6.2531	6.1982	6.3588	6.2987	6.3072
mass change (%)	−0.052	−0.043	−0.064	−0.64	−0.48	−0.79	0.0048	0.0096	0.0081	−0.058	−0.035	−0.062

**Table 5 materials-18-01328-t005:** Elemental composition of corrosion products in the cross-section of AFA2-C-Y (wt.%).

Elements	O	Al	Cr	Fe	Ni	Pb	Bi
1	17.78	3.18	15.58	35.51	23.93	0	0
2	22.78	4.5	18.02	33.33	16.17	0	0
3	29.93	0.02	0.85	66.02	0.47	0	0
4	0.79	2.65	12.25	63.54	16.89	0	0
5	26.57	5.03	21.52	28.54	13.43	0	0

**Table 6 materials-18-01328-t006:** Elemental composition of corrosion products in the cross-section of AFA1-C-MnY (wt.%).

Elements	O	Al	Cr	Fe	Ni	Pb	Bi
1	17.78	3.18	15.58	35.51	23.93	0	0
2	22.78	4.5	18.02	33.33	16.17	0	0
3	29.93	0.02	0.85	66.02	0.47	0	0
4	0.79	2.65	12.25	63.54	16.89	0	0
5	26.57	5.03	21.52	28.54	13.43	0	0

**Table 7 materials-18-01328-t007:** Elemental composition of corrosion products in the cross-section of AFA3-C-Mn (wt.%).

Elements	O	Al	Cr	Mn	Fe	Ni	Pb	Bi
1	20.02	2.45	19.17	9.53	29.71	3.83	5.48	6.63
2	19.17	8.02	13.52	4.81	42.12	6.12	3.19	0.06
3	1.34	2.62	12.2	4.06	59.99	16.08	0	0.2

**Table 8 materials-18-01328-t008:** Elemental composition analysis of oxidation region of AFA3-C-Mn cross-section (wt.%).

Elements	O	Al	Cr	Mn	Fe	Ni	Nb	Mo	Pb	Bi
1	1.41	0.02	0.26	0.48	1.6	1.51	0.21	1.01	ND	93.51
2	37.08	ND	41.92	16.96	1.44	0.33	0.24	0.17	1.5	0.31
3	1.67	0.19	5.83	0.22	87.89	2.8	ND	1.39	ND	ND
4	1.07	0.1	5.42	0.12	88.67	3.51	ND	1.08	0.01	ND
5	32.15	0.12	30.21	13.55	10.24	1.16	0.09	1.27	9.55	1.66
6	44.31	9.56	14.47	28.09	1.67	0.36	ND	0.93	0.27	0.34
7	85.51	0.09	1.48	0.25	3.64	2.11	0.69	0.99	2.79	2.15
8	4.83	0.05	1.49	0.77	84.23	5.84	ND	2.48	0.31	ND
9	38.32	ND	28.08	19.45	11.06	0.29	ND	2.5	ND	0.31
10	37.15	3.26	32.16	18.88	1.87	0.47	0.73	4.77	ND	0.71

## Data Availability

The original contributions presented in this study are included in the article. Further inquiries can be directed to the corresponding authors.

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
