# Peer review of "Corrosion Resistance of Mn-Containing AFA Alloys in LBE at 550 °C with Controlled Oxygen Concentration of 10−6 wt.%"

_materials, 2025, doi:10.3390/ma18061328_

Round 1
Reviewer 1 Report
Comments and Suggestions for Authors
The corrosion resistance of new AFA steels is presented. It is therefore an experimental study that is presented: corrosion tests at only one temperature in liquid LBE and SEM analyses and TEM analyses for one case. No XRD analysis was carried out, contrary to what is usually done. The interpretation of the results is based on a discussion that is not supported by references to the bibliography.
The work and so the article need to be improved.
Introduction: the authors say that few studies on the corrosion of AFA steels have been conducted: this is partly true. But the authors present too few articles compared to all those that have been published. Moreover, the authors never compare their results with those in the literature (even if they are not numerous).
Experiment:
The presentation of the microstructure of the materials, in particular the grain size, are missing. Corrosion resistance depends among other things on grain size. Grain boundaries are diffusion paths. This aspect is never discussed or mentioned.
Corrosion tests need to be more described. How many tests per material? In mass loss table, there is no standard deviation. Does this mean that there was only one test per condition and that everything is based on one test.
Results
Element composition at different points are presented in the artiche for differents cases : Why aren't there pictures with the location of the points?
Discussion:
What real informations does Figure 10 give?
Comments on the Quality of English Language
To improve
Author Response
Thank you very much for your valuable suggestions. We have addressed your comments and provided detailed responses in our reply document. If there are any areas where our understanding or revisions are still insufficient, please do not hesitate to point them out. We will make every effort to further improve our manuscript.

Reviewer 2 Report
Comments and Suggestions for Authors
Corrosion behaviour of alumina-forming austenitic steels (AFA steels) containing Mn are in the interest of the authors of the paper. This study investigates the corrosion resistance of Mn-containing AFA steels in Lead-Bismuth Eutectic (LBE) at 550 °C with a controlled oxygen concentration of 10⁻⁶ wt%.
By analysing literature, authors of the paper decided to perform >>research on the application of AFA alloys in lead-cooled reactors remains relatively limited<<. It is known that the oxidation mechanism of oxygen-controlled LBE is similar to that of metal oxidation in other oxygen-containing environments.
After Introduction, the authors presented method and materials, with experimental results displayed in 15 Figures and 4 Tables. Under the studies, corrosion behaviour of four different AFA steel alloys at 550 °C in 10-6 wt% oxygen-controlled LBE was analyzed. The time of 1000 h was assumed to exposure the samples for corrosion and compare the results obtained. A discussion of experimental results were done to formulate concusions. AFA steels with the addition of Mn were investigated to find their corrosion resistance. Finally, there are four conclusions presented.
The manuscript is written using appropriate language; there are some points to correct (e.g. >>wt.%<<, verse 34) as indicated in the PDF file with color highlights. Lacking units on vertical axis in Figures 5a, 7c, 8b, 14b should be added.
Please refer to the attachment for additional comments.

Author Response
Thank you very much for your insightful comments on our manuscript. We have fully accepted your suggestions and have made the necessary revisions accordingly. The detailed changes can be found in our reply document as well as in the revised manuscript.

Reviewer 3 Report
Comments and Suggestions for Authors
The authors are advised to revise the manuscript and address the following comments to enhance its quality:
- Lines 97–113: Remove bullet points and replace them with a table for process parameters.
- Table 2: The authors should add the variation in composition as well.
- Table 3: Add standard deviation for experimental observations. The mass change reported here may not be insignificant when standard deviation is included in the results.
- Lines 200–208: Bullet points are unsuitable for standard journal papers.
- Figure 8: What is the reason for the variation in the position of the change in composition of each element at the interface? Is it related to changes in diffusion kinetics or corrosion rates?
Round 2
Reviewer 1 Report
Comments and Suggestions for Authors
Very few evolutions to the text have been made in relation to the remarks.
Too few references are still cited concerning AFA steels (and a so-called review reference (which is not one!) cannot justify citing only a few articles) and there is no real discussion of the results with respect to the literature.
The question regarding the presence or absence of standard deviation does not concern the precision of the balance (which we hope correct) but the number of corrosion tests per conditions. How many samples were tested and studied for the same conditions?
The article focuses on the formation of surface oxides, a form of corrosion. But this depends at high temperature on diffusion phenomena and therefore on the microstructure (especially presence of precipitates) and grain boundaries. Thus, the authors cannot fail to give data on the microstructure and grain sizes ....
Author Response
I sincerely apologize for misunderstanding your comments during the previous revision process. Your feedback has been immensely helpful, and I have carefully revised the manuscript again based on your valuable suggestions. Please refer to the response letter and the revised manuscript for detailed changes.

Round 3
Reviewer 1 Report
Comments and Suggestions for Authors
I have read your answers and the few changes you have done. They do not seem sufficient to me because they do not answer key points and questions.
The references cited are too few in number compared to the studies carried out these last years. Furthermore, in the discussion your results are not discussed enough in relation to those in the literature.
The question regarding the standard deviation was not about the precision of the measurement since we assume that before publishing, the authors check the accuracy of the means they use but about the number of samples tested and studied for each condition. How many samples per condition? Are the results reproducible (between two samples)? With what standard deviation on the measurements?
You justify not giving the grain size and informations on the microstructure of the materials by stating that you are only interested in the oxides that form. It is known that, particularly in temperature, the formation of oxides is linked to the diffusion paths of the elements, their presence in the microstructure and therefore to the presence of grain boundaries and/or precipitates. This is why, for me, it is nonsense to study the presence of oxide, and explain these formation, without presenting the microstructure of the material (including the grain size) and without this being discussed in the discussion.
Author Response
Dear Reviewer,
Thank you very much for your valuable comments. We have taken your suggestions very seriously and made the following improvements in our manuscript:
Additional References: We have added more up-to-date references in the discussion section to enrich the research background.
Data Enhancement: We have incorporated the standard deviation in the corrosion weight gain section to more accurately reflect the reliability of our experimental data.
Regarding the issue of grain size and grain boundaries that you raised, the focus of this study is on the corrosion behavior of Mn-containing AFA steel in liquid lead-bismuth, with particular emphasis on the formation of the oxide film. Our previous research has shown that the formation of the oxide film is the most critical factor influencing corrosion resistance in liquid lead-bismuth environments. Therefore, this paper is centered around this phenomenon. The four conclusions we have drawn are also unrelated to diffusion or changes in grain size, as they are primarily summaries of the experimental observations.
We plan to address the corrosion mechanisms, including the effects of grain size and grain boundaries, in a subsequent publication.
Thank you again for your understanding and support. We believe that these improvements have enhanced the scientific rigor and completeness of our manuscript.
Best regards,
Menghe TU

Round 4
Reviewer 1 Report
Comments and Suggestions for Authors
Add more references